# Immunohistochemical Biomarkers as a Surrogate of Molecular Analysis in Ovarian Carcinomas: A Review of the Literature

**DOI:** 10.3390/diagnostics11020199

**Published:** 2021-01-29

**Authors:** Giacomo Santandrea, Simonetta Piana, Riccardo Valli, Magda Zanelli, Elisa Gasparini, Antonio De Leo, Vincenzo Dario Mandato, Andrea Palicelli

**Affiliations:** 1Clinical and Experimental Medicine PhD Program, University of Modena and Reggio Emilia, 41121 Modena, Italy; 2Pathology Unit, AUSL-IRCCS Reggio Emilia, 42123 Reggio Emilia, Italy; Simonetta.piana@ausl.re.it (S.P.); Riccardo.Valli@ausl.re.it (R.V.); Magda.Zanelli@ausl.re.it (M.Z.); Andrea.Palicelli@ausl.re.it (A.P.); 3Oncology Unit, AUSL-IRCCS Reggio Emilia, 42123 Reggio Emilia, Italy; Elisa.Gasparini2@ausl.re.it; 4Molecular Diagnostic Unit, AUSL Bologna, Department of Experimental, Diagnostic and Specialty Medicine, University of Bologna, 40138 Bologna, Italy; antonio.deleo@unibo.it; 5Unit of Obstetrics and Gynaecology, AUSL-IRCCS Reggio Emilia, 42123 Reggio Emilia, Italy; vincenzodario.mandato@ausl.re.it

**Keywords:** ovarian cancer, biomarkers, molecular classification, immunohistochemistry

## Abstract

The term “ovarian carcinoma” encompasses at least five different malignant neoplasms: high-grade serous carcinoma, low-grade serous carcinoma, endometrioid carcinoma, mucinous carcinoma, and clear cell carcinoma. These five histotypes demonstrated distinctive histological, molecular, and clinical features. The rise of novel target therapies and of a tailored oncological approach has demanded an integrated multidisciplinary approach in the setting of ovarian carcinoma. The need to implement a molecular-based classification in the worldwide diagnostic and therapeutic setting of ovarian cancer demanded a search for easy-to-use and cost-effective molecular-surrogate biomarkers, relying particularly on immunohistochemical analysis. The present review focuses on the role of immunohistochemistry as a surrogate of molecular analysis in the everyday diagnostic approach to ovarian carcinomas.

## 1. Introduction

Ovarian epithelial cancer (OC) is the eighth most common malignant neoplasm in female patients and the seventh greatest cause of cancer death worldwide [1,2,3,4,5,6,7]. Although several questions regarding the molecular heterogeneity and the pathogenesis of OC still remain open, in the past century great efforts have been devoted to understand the complex landscape of OC. In 2010, Kurman and Shih proposed the first attempt at a molecular classification of OCs, which were divided in two main groups: (1) Type I carcinomas, comprising low-grade serous carcinoma (LGSC), low-grade endometrioid ovarian carcinomas (ENOC), clear cell carcinomas (CCC), mucinous carcinomas (MC) and malignant Brenner tumors; (2) Type II carcinomas, including high-grade serous carcinomas (HGSC), high-grade ENOCs, undifferentiated carcinomas, and carcinosarcomas. Type I carcinomas were generally associated with a good prognosis, while Type II carcinomas, typically *TP53*-mutated, were characterized by aggressive clinical behavior and poor outcomes [8,9]. From this dualistic classification, the introduction of new molecular technologies expanded our knowledge of OCs and demonstrated at least five main entities with independent histological, clinicopathological, and molecular features: HGSCs, LGSCs, ENOCs, MCs, and CCCs. The Cancer Genome Atlas Network (TCGA) studies on HGSC and endometrial carcinomas pushed our knowledge even further [10,11]. Although still in an early phase of development, the need for a molecular-based classification is growing, in order to better stratify and develop target tailored therapies for OC patients. This molecular-based approach has notable limitations, in particular regarding cost-effectiveness; therefore the search for an easily available and cost-effective surrogate for molecular analysis shifted to alternative methods such as immunohistochemistry (IHC) and traditional targeted DNA sequencing. The aim of the present study is to review recent evidence on the molecular biology of OCs, focusing on the development of immunohistochemical biomarkers as surrogates for molecular classification. 

## 2. High-Grade Serous Carcinoma

HGSC is the most common epithelial malignancy of the tubo-ovarian district, responsible for almost 70% of OCs [2]. The majority of HGSCs present at advanced clinical stage and the overall five-year survival is poor, ranging from 10 to 40% [1]. Currently, two main subtypes of HGSC are recognized on histologic examination: Classic histotype: characterized by a papillary, micropapillary, and/or solid architecture, marked nuclear pleomorphism and high mitotic index (Figure 1a).“SET” variant (solid-pseudoendometrioid and transitional): characterized by an admixture of solid, glandular/endometrioid-like, and transitional/malignant Brenner-like growth patterns), higher mitotic index compared to the classic histotype, and a high number of tumor-infiltrating lymphocytes (TILs) (Figure 1b). In 2012, a study from Soslow et al. demonstrated a statistical association between *BRCA1/2* mutation and SET morphology [12]

The TCGA project revealed a surprisingly simple spectrum of mutations in HGSC [10]. *TP53* mutation has been observed in almost all tubo-ovarian HGSCs (96%), followed by alteration of homologous recombination repair-related genes such as *BRCA*1/2. 

The use of an optimized p53 IHC staining as a molecular surrogate for *TP53* mutations was investigated by Köbel et al., 2016 [13,14]. An almost perfect correlation between p53 IHC and *TP53* mutation was obtained when a binary p53 IHC scoring system was adopted: p53 intense nuclear positivity in >80% tumor cells (overexpression pattern), complete absence of expression (null pattern) or cytoplasmic expression without nuclear staining were scored as p53 “abnormal” (p53abn) and correlated with TP53 mutation (Figure 2a,b).heterogeneous p53 expression was scored as “wild-type” (p53wt) and correlated with a *TP53*-wildtype gene (Figure 2c).

Though p53 mutational pattern is now considered a reliable marker for a diagnosis of HGSC, two important caveats have to be considered: (1) 4% of HGSCs are p53 wild-type; (2) almost 20% of ovarian endometrioid grade 3 carcinomas (morphologically similar to HGSC) and a minor subset of CCC are p53abn [15,16]. *TP53*-wildtype HGSC, albeit rarely encountered, can represent a potential diagnostic pitfall, as grade 3 endometrioid carcinoma, often *TP53*-wildtype, can mimic a SET variant of HGSC. The following criteria could help in the differential diagnosis:Association with serous tubal intraepithelial carcinoma (STIC), low-grade serous-like areas, WT1 IHC positivity, mutation of *CCNE1*, *BRCA1/2*, and *MDM2* amplification support the diagnosis of *TP53*-wildtype HGSC [17].Association with endometriosis, endometrioid cystadenofibroma, and borderline endometrioid tumor, as well as WT1 IHC negativity, support the diagnosis of p53abn grade 3 ENOC [18].

The adoption of prophylactic salpingo-oophorectomy in patients with *BRCA* mutation and extensive adnexal sampling via the so-called SEE-Fim protocol (sectioning and extensively examining the FIMbriated end) provided a basis to identify potential early lesions in the tubo-ovarian district [19]. This resulted in the discovery of the distal fimbria epithelium as a putative precursor for HGSC development via early *TP53* mutation [20,21,22,23]. A combined morphologic-immunohistochemical analysis, based on the degree of nuclear atypia, p53 expression, and Ki-67 index, was proposed to identify a spectrum of tubal lesions named p53 signature, serous tubal intraepithelial lesion (STIL) and STIC (Figure 3) [23,24]. Though reporting p53 signature and STIL is not considered of clinical interest, the last *WHO classification of the Tumours of the Female Genital Tract* and the *AJCC Cancer Staging Manual* encouraged thorough examination of the fimbria to search for STIC [1,25]. Albeit STIC is an intraepithelial lesion, it demonstrated a peritoneal metastatic potential; thus, STIC should not be considered a precancerous lesion, but a de facto early-stage HGSC.

*BRCA1* and *BRCA2* are tumor suppressor genes involved in the so-called homologous recombination repair system (HR). In the event of DNA double-strand breaks or DNA damage at a replication fork, the cells undergo a repair via HR, recruiting the sister chromatid as a template for DNA replication [26]. In particular, *BRCA*1 together with BARD1 and BRIP1 recruit the so-called MRN complex (MRE11, RAD50, and NBS1) at the site of the damage. The MRN complex then resects the 5′ strand, while *BRCA*2 is responsible for loading RAD51 protein at the site of the single-strand DNA stretch to initiate the homologous DNA invasion [27]. In the event of HR deficiency, the damaged cell undergoes DNA repair via non-homologous end joining, a more error-prone DNA repair system, leading to genetic instability. *BRCA1/2* germinal loss of function is the driver mutation of the so-called hereditary breast and ovarian carcinoma syndrome (HBOC). HGSCs are the most commonly found ovarian histotype in HBOC patients.

Considering germline *BRCA1* and *BRCA2* pathogenic variants (8% and 9% of the cases respectively), *BRCA1* and *BRCA2* somatic mutations (3% overall), and other alterations such as hypermethylation of *BRCA1* promoter and mutations in other HR-related genes such as *RAD51C*, *ATM*, *PALB2*, *CDK12*, *CHEK2*, etc., almost 50% of HGSCs could be classified as homologous recombination deficient (HRD) [28,29,30,31]. 

In a study by Meisel et al., immunohistochemistry for *BRCA*1/2 as a surrogate biomarker for *BRCA* mutation showed high sensitivity (86.2%) and negative predictive value (95.4%). However, it demonstrated poor specificity (78.3%) and positive predictive value (52.1%), therefore IHC-only testing should not be considered sufficient for the detection of *BRCA* genes status [32,33]. Homologous recombination repair functionality has been tested in breast and endometrial carcinoma via immunofluorescence with excellent results, making it a promising alternative tool to identify HRD tumors. However, no immunofluorescence HRD-test has been carried on OCs to date: further studies are therefore needed to routinely implement HRD testing on OC [34,35,36,37]. 

The adoption of a routine workflow to identify *BRCA1/2* mutations has been highlighted with the introduction of poly(ADP-ribose) polymerase inhibitors (PARPi) as maintenance therapy in platinum-sensitive OCs [38,39]. PARPi selectively inhibit the enzyme poly(ADP-ribose) polymerase (PARP), involved in DNA single-strand break repair. In the event of *BRCA1/2* loss, tumor cells are unable to undergo DNA repair via HR and single-strand break repair, causing irreparable damage and, ultimately, cell death. Several clinical trials, after the milestone SOLO1 trial, demonstrated a statistically significant better outcome in *BRCA* germline-mutated patients [40]. However, to expand the potential number of patients that could benefit from a PARPi treatment, several ongoing studies are focusing on the outcome of *BRCA* somatic tumors and/or HRD non-*BRCA* patients [41,42,43,44,45]. 

Thus a pathology-based, *BRCA*-reflex testing seems to be particularly useful to:identify patients with *BRCA* germline mutations eligible for PARPi therapy, prophylactic surgery, and genetic counseling;identify patients with *BRCA* somatic mutations potentially eligible for PARPi therapy;exclude unnecessary germline testing for somatic *BRCA*-negative tumors, in order to be cost-effective and reduce patients’ psychological distress.

The role of immunotherapy in OC is still controversial as the response rate to checkpoint inhibitors in unselected OCs is modest as demonstrated by the KEYNOTE-100 trial [45]. In HGSCs, DNA damage, induced by platinum and PARPi therapy, has been suggested to increase the production of tumor cell neoantigens, highlighting a potential role of checkpoint inhibitors in combination with PARPi [46]. Furthermore, higher expression of PD-1/PD-L1 by immune tumor environment, and a higher number of CD3+ and CD8+ lymphocytes have been associated with better prognosis in HGSCs [47,48,49]. According to Chen et al., PD-L1 expression among different histotypes was highest in HGSCs: almost 50% of HGSCs demonstrated a >1% combined positive score (CPS) (evaluated in tumor and immune cells) compared to 21% of HGSCs revealing a >1% tumor proportion score (TPS) (evaluated only in tumor cells) [49,50]. Noteworthy PD-L1 positivity was not associated with *BRCA* status, but demonstrated a statistically significant correlation with better survival in late-stage HGSCs (Table 1) [50].

## 3. Low-Grade Serous Carcinoma

Ovarian LGSC represents an uncommon epithelial neoplasm (5% of all OCs) [1,2,51]. LGSC is commonly associated with serous cystoadenofibromas and/or serous borderline tumors. Tumor histology is characterized by micropapillary or cribriform architecture with bland nuclear atypia and low-mitotic count (Figure 4). The IHC profile is typically PAX8+, WT1+, and p53wt: the latter phenotype is particularly helpful in the differential diagnosis between LGSC and classic HGSC. Early-stage LGSCs have an excellent prognosis, while stage III-IV tumors have poor prognosis, mainly due to poor response to conventional chemotherapy.

Despite their similar pathological denomination, LGSCs share little in common with HGSCs. HGSCs are thought to arise from the surface epithelium of the distal fimbria via *TP53* and HR-related gene mutations, while LGSC is thought to be the result of a neoplastic transformation of cortical inclusion cysts into serous cystadenomas/adenofibromas and serous borderline tumors [2]. Furthermore, the evidence of an evolution from LGSC to HGSC is extremely rare. 

LGSCs are characterized by a completely different mutational spectrum compared to HGSCs: mutations in *KRAS*, *NRAS*, *BRAF*^V600E^, *BRAF*^non-V600E^, and *ERBB2* were found in almost 50% of LGSCs, while *TP53* was wild-type in all reported cases [52,53,54,55,56]. *KRAS* p.Gly12Val (*KRAS*^G12V^) mutation has been particularly associated with worse prognosis and higher recurrence rate, while *BRAF*^V600E^ demonstrated a statistically significant correlation with better prognosis [55].

To further complicate the landscape of serous carcinomas, Zarei et al. recently published a case series of serous carcinomas with mixed low-grade and high-grade features [56]. This evidence is particularly of interest as this entity seems to question the apparent dualistic nature of serous carcinomas. These tumors showed a relative rarity of mutations typical of HGSC and LGSC and a particularly aggressive clinical behavior. This suggests a potential alternative oncogenic pathway, although further studies are needed to better characterize these rarely encountered entities.

Hormonal replacement therapy is currently included in the National Comprehensive Cancer Network (NCCN) guidelines in advanced-stage LGSC with expression of estrogen receptor (ER) and progesterone receptor (PGR) [57]. Some studies focused on LGSCs’ response to *BRAF* inhibitors, with promising results: this evidence is particularly noteworthy as LGSC is a stage-dependent disease, poorly responding to conventional chemotherapy in advanced stages [58,59,60,61]. Currently, Trametinib is contemplated as a treatment of advanced-stage LGSC in the NCCN guidelines [57]. In a study by Turashvili et al., IHC for *BRAF*^V600E^ demonstrated 96% sensitivity, 96% specificity, 93% positive predictive value, and 98% negative predictive value for *BRAF*^V600E^ mutation [60]. Thus, IHC evaluation of *BRAF*^V600E^ is particularly helpful and relatively cost-effective for the identification of patients eligible for target therapy, especially in advanced-stage LGSC. 

To the authors’ knowledge, no studies have been reported to date on a potential therapeutic role of anti-HER2 targeted drugs in *ERBB2*-mutated LGSCs (Table 2).

### 3.1. Ovarian Endometrioid Carcinoma

Ovarian endometrioid carcinomas (ENOCs) represent a heterogeneous group of neoplasms regarding morphological and molecular features. ENOCs typically exhibit cribriform, maze-like or glandular architecture with frequent squamous differentiation [1,2]. Tumor histological grade is based on the percentage of the solid non-squamous component of the tumor, as for its endometrial counterpart, according to the FIGO grading. In particular, FIGO grade 1, 2, and 3 tumors have ≤5%, 6–50% and >50% of solid, non-squamous, growth pattern: marked nuclear atypia in >50% of tumor cells may upgrade the tumor of 1 grade (Figure 5a,b). ENOCs may arise in association with endometrioid cystadenofibroma and endometrioid borderline tumor and/or in the context of pelvic endometriosis [1]. The typical immunoprofile is PAX8+, WT1-, ER+, PGR+. Prognosis is stage-dependent, with excellent outcomes in stage I-II tumors and poor outcomes in advanced stages [61].

The 2013 TCGA analysis of endometrial carcinomas (ECs) identified four main molecular subgroups: DNA polymerase ε (*POLE*) ultramutated, microsatellite instable (MSI-H), carcinomas with high somatic copy number alterations (CN), and carcinomas with low CN [11]. These four subgroups demonstrated different mutational spectra and biological behavior. *POLE* is a gene involved in DNA repair and replication. The exonuclease proofreading activity replaces erroneously incorporated nucleotides during DNA replication. Mutations in its exonuclease domain lead to proofreading errors, with increase in mutation rates during DNA replications by about 100-fold, and, ultimately, neoplastic transformation [62,63,64]. Mismatch repair (MMR) proteins (*MLH1*, *PMS2*, *MSH2*, and *MSH6*) are nuclear enzymes involved in repair of base–base mismatch during DNA replication in proliferating cells: these proteins form heterodimers binding to areas of abnormal DNA and initiating its removal. Loss of MMR proteins leads to accumulated errors during DNA replication, especially in short repetitive nucleotide sequences (“microsatellite instability”). Mutation of MMR-genes and hypermethilation of MLH1 promoter has been associated with the so-called MSI-H group and, for germline mutations, with Lynch Syndrome (LS) [65,66,67]. CN^high^ ECs exhibited frequent mutation in *TP53*. CN^low^ ECs were characterized by an heterogeneous spectrum of mutated genes such as *PTEN*, *ARID1A*, *CTNNB1*, and *PIK3CA* [68]. 

Interestingly, endometrial endometrioid carcinomas (EECs) were represented in all four subgroups. In particular, low-grade EECs were most frequently encountered in the CN^low^ and MSI subgroups while high-grade EECs were almost equally distributed in all four molecular subgroups [69,70]. This evidence has notable clinical implications; in fact, *POLE* ultramutated carcinomas demonstrated an excellent prognosis, MSI-H and CN^low^ ECs showed intermediate prognosis, while the CN^high^ group was associated with worse prognosis [11]. As to this promising prognostic stratification of EC-patients, the 2020 *WHO classification of the Tumours of the Female Genital Tract* tried to implement the usual histological classification with a new molecular approach in pathologists’ routine practice: a diagnostic algorithm (using Sanger sequencing and immunohistochemistry as cheaper surrogates of the expensive molecular analysis) was proposed to identify molecular EC-subgroups (Table 3).

Immunohistochemistry for MMR proteins and p53 are excellent surrogates for MSI-H and CN^high^ tumors. No immunohistochemical marker is available for *POLE*, thus Sanger sequencing is the gold standard to evaluate *POLE* status. Finally, the assignment to the CN^low^/no special molecular profile (NSMP) group could be made once *POLE*/MMR/p53 mutations are excluded. 

Recently, ENOCs demonstrated the same characteristics of EECs regarding histological, clinical, and molecular parameters [16]. The four molecular subgroups of ENOC were classified according to the 2013 TCGA classification of EECs and using an algorithm which prioritized *POLE* sequencing, with subsequent MMR and p53 immunohistochemical evaluation of *POLE*-wt cases (Figure 6). 

*POLE* ultramutated ENOCs, albeit rarer than their endometrial counterpart, demonstrated the same excellent prognosis. Mismatch repair deficient (MMRd) ENOCs were also rarer than MMRd EECs, while NSMP ENOCs were more frequently represented than NSMP EECs: both MMRd ENOCs and NSMP ENOCs were associated with an intermediate prognosis. Finally, p53abn tumors, rarely encountered, were significantly correlated with poor prognosis. 

In another study by Parra-Hernan et al., MMRd ENOCs demonstrated excellent prognosis, overlapping with *POLE*-mutant ENOCs [69]. These molecular subgroups demonstrated a better correlation with tumor prognosis than the traditional morphologic evaluation of tumor grade. As an example, low-grade p53abn ENOCs showed aggressive clinical behavior, contrariwise high-grade *POLE* ENOCs were associated with an excellent prognosis. Thus, the role of molecular classification in ENOCs appears promising as in EECs, especially to guide neoadjuvant treatment.

Histopathological and IHC analysis, blinded of the molecular status, was carried on a series of ECs to identify morphological features associated with *POLE* mutations by Van Gool et al. [70]. *POLE*-mutant ECs were statistically associated with a high number of TILs, tumor giant-cells, p53wt IHC pattern, and intact MMR IHC expression. The same morphological features were also observed in *POLE* ENOCs [71]. Therefore, a simple morphological and IHC approach could identify patients with features suggesting *POLE* mutation, albeit Sanger sequencing represents the gold standard assay to establish *POLE* mutational status. 

A large cohort of 502 early-stage ovarian carcinomas, representative of all five main histotypes was tested for MMRd by Leskela et al. MMRd was significantly associated with endometrioid and clear cell histotypes [72]. As for its colorectal and endometrial counterparts, the *MLH1* promoter hypermethylation was the most commonly found alteration (86% of the cases), but germline loss of function of *MLH1*, *MSH2*, and *MSH6* (highly suspicious for LS) was observed in 35% of the cases. Furthermore, MMRd OCs were associated with younger age at presentation and increased TILs. A two-antibody approach with PMS2 and MSH6 is considered sufficient and more cost-effective than an upfront four-antibody approach, due to the binding heterodimeric nature of the MMR complexes: a loss of expression of MLH1 or MSH2 would inevitably lead to degradation of PMS2 and MSH6, respectively [73]. Conversely, a loss of expression of PMS2 and MSH6 would not lead to a degradation of MLH1 and MSH2 [74,75,76]. However, in the event of a PMS2 and MSH6 loss, confirmatory IHC for MLH1 and MSH2 should be performed [76]. An *MLH1* promoter methylation test should also be carried out in all cases showing MLH1 IHC loss.

PD1/PD-L1 expression has been tested in OC with particular attention to MMRd and *POLE*-mutant tumors, because of their high immunogenic characteristics [3,48,77,78]. This evidence is of particular interest for future therapeutic implications. The current NCCN guidelines contemplate the use of Pembrolizumab (anti-PD1) for MSI-H carcinomas [57].

Hormone receptor expression has been linked to better prognosis supporting the possibility of hormonal therapy in ENOCs. In a large multicentric study by Sieh et al., ER and PGR expression, scored as negative (<1% positive nuclei), weak (1–50%), or strong (≥50%), were correlated with better disease-specific survival in ENOCs [79]. Another study from Rambau et al. confirmed a statistically significant better outcome in hormone-positive ENOCs [80].

IHC evaluation of PTEN and ARID1A proteins are good surrogates of loss-of-function mutations in *PTEN* and *ARID1A* genes. *PTEN* loss could represent a target for PI3K/AKT inhibitors, while ARID1A loss for EZH2 and HDAC inhibition, but further studies are needed to clarify their potential role [81,82,83].

*CTNNB1*-mutated ENOCs, examined with surrogate IHC staining for nuclear β-catenin, were associated with a favorable outcome in two independent studies, while abnormal expression of p16 in ENOCs has been associated with a worse outcome: however, these data still need further validation (Table 4) [84,85,86].

To summarize, the spectrum of mutations in ENOCs is highly variable and data are still limited. The overall prognosis of ENOCs is good but the molecular subgrouping via IHC surrogate biomarkers could drive the choice and refine the adjuvant treatment guidelines shortly.

### 3.2. Clear Cell Carcinoma

Ovarian CCCs represent 10% of ovarian malignant neoplasms and typically present at early-stage. The strongest prognostic factor in CCCs is the clinical stage, with an overall good prognosis in stages I-II and poor prognosis in stages III-IV [87]. Histologically, CCCs are characterized by a mixture of tubulocystic and papillary epithelial structures associated with a distinctive hyaline, acellular stroma (Figure 7a,b) [1]. Tumor cells are cytologically bland, with a low mitotic count and often subtle invasion. Hobnail clear cells are typical findings although an oxyphilic variant of CCC has been described [88,89,90,91]. The IHC profile is typically PAX8+, napsin A+, hepatocyte nuclear factor 1β+, WT1-, ER-, PGR-. Tumors are typically associated with endometriosis and clear cell borderline and/or adenofibroma components. Notably, tumors associated with endometriosis or adenofibromatous component demonstrated a better prognosis than *de novo* CCCs. 

The mutational spectrum of CCCs is characterized by frequent mutation of SWI/SNF chromatin remodeling complex protein *ARID1A*, loss of *PTEN*, activation of *PIK3CA*, *TERT* promoter mutations and, rarely, *TP53* mutation (5%) [83,92,93,94,95]. Notably, almost 10% of ovarian CCCs have been linked to LS; therefore MMR testing is routinely recommended in all CCCs [96]. This is of particular significance as Lynch-associated CCCs, even at an advanced stage, demonstrated better prognosis, possibly due to high tumor immunogenicity and high PD-1/PD-L1 expression, similarly to ENOCs [48,97].

### 3.3. Mucinous Carcinoma

Primary ovarian mucinous carcinoma is extremely rare as the vast majority of mucinous malignant tumors represents metastatic localization from the cervical or gastrointestinal tract [98]. Commonly accepted diagnostic clues, favoring an ovarian origin, are large unilateral localization, size >10 cm, and association with mature cystic teratoma, Brenner’s tumor, mucinous cystadenoma or mucinous borderline tumor. Metastatic mucinous carcinomas can be deceptively bland, mimicking a primary ovarian borderline mucinous tumor [99,100,101]. Immunohistochemistry could aid the differential diagnosis, as ovarian MCs are cytokeratin 7 (CK7) positive in >80% cases, only focally and weakly positive for CK20 and CDX2, usually negative for SATB2 [102]. Conversely, tumors of colorectal origin are usually CK7-/CK20+/CDX2+ and SATB2+. The differential diagnosis could be particularly challenging with an upper-GI primary: careful clinical and radiological examination is therefore mandatory. Prognosis is strictly stage-dependent with a reported 75–90% five-year survival rate for stage I/II MCs versus 17% for stage III-IV MCs [1].

Molecularly, *CDKN2A* and *KRAS* mutations are the most commonly encountered followed by *TP53* and *ERBB2* mutations [103,104]. MSI was found in almost 20% of the cases [96]. *ERBB2* amplified tumors demonstrated a significantly better prognosis when Trastuzumab was added to the conventional chemotherapy treatment, suggesting a potential actionable target in the treatment of ovarian primary MCs [104]. 

### 3.4. Other Rare Primary Ovarian Tumors

Seromucinous carcinomas (SCs), previously considered an independent entity, are now considered a subtype of endometrioid carcinoma due to similar pathogenic background (endometriosis-associated) and mutational spectrum: thus, the last WHO Classification of Female Genital Tract Tumors considers SCs as a morphologic variant of ENOCs [105,106]. 

A new entity introduced in the 2020 WHO Classification of Female Genital Tract Tumors is the mesonephric-like adenocarcinoma (MLA). This rare tumor may arise from a malignant transformation of mesonephric ovarian remnants or transdifferentiation of Müllerian epithelium. The latter event could explain the frequent association of MLA with endometriosis or other carcinomatous histotypes, most commonly ENOCs. MLAs usually exhibit an admixture of glomeruloid, tubular, sieve-like, solid, and glandular architectural patterns and a distinctive GATA3+/TTF1+/CD10+(luminal)/PAX8+ immunophenotype, while ERs and PGRs are typically negative [107]. Though extremely rare, these neoplasms demonstrated distinctive molecular characteristics, such as *KRAS*, *NRAS*, and *PIK3CA* mutations [108,109]. No prognostic data are available due to the rarity of these tumors.

Dedifferentiated/undifferentiated carcinomas (DED/UCs) are uncommon ovarian tumors with poor prognosis, thought to represent the latter stage of neoplastic transformation of endometrioid tumors. Frequent mutations in the SWI/SNF chromatin remodeling complex have been found, in particular involving *ARID1A*, *ARID1B*, *SMARCB1/INI1*, and *SMARCA4* genes [1]. Notably, a subset of DED/UC is characterized by MSI and *POLE* EDM. All these molecular alterations can be also easily identified via IHC and sequencing with good reliability and can be of help in the differential diagnosis with other undifferentiated metastatic neoplasms. This can also be helpful to identify a subset of patients potentially eligible for immunotherapy, as demonstrated for their endometrial counterparts. In a case series published by Espinosa et al., a neuroendocrine differentiation was encountered in four ovarian DED/UCs, one of which, characterized by *POLE* EDM, demonstrated excellent prognosis [110].

Carcinosarcomas (CSs) are rare biphasic ovarian malignant tumors with both carcinomatous and sarcomatous elements [1]. The two components seem to be of epithelial origin as the sarcomatous areas have been demonstrated to derive from an epithelial-mesenchymal transition [111]. Typically diagnosed at a late clinical stage, CSs are resistant to conventional chemotherapy and usually have a poor prognosis [112]. The molecular landscape of CSs is similar to their uterine counterpart with recurrent mutations in *TP53*, *PIK3CA*, *BRCA1/2*, *PTEN*, *MLH1*, *MSH6*, *PPP2R1A*, *ARID1A*, *KRAS*, *CDH4*, and *BCOR* [113,114,115,116]. Actionable mutations are represented by *BRCA1/2*: these tumors can be potentially treated with PARPi. In fact, as for HGSCs, CSs with *PTEN* loss could be eligible for PI3K/AKT inhibitors and, although rarely encountered, microsatellite instable-CSs could be eligible for anti-PD1 therapy [117].

## 4. Conclusions

OCs are a heterogeneous group of diseases with specific histologic, molecular, and clinicopathologic features. Ten years after the first attempt at an OC molecular classification, the introduction of new technologies allowed us to expand our knowledge of the complex molecular landscape of this disease. To date, at least five main independent histotypes are recognized: HGSCs, LGSCs, CCCs, ENOCs, and MCs [2,8]. A molecular subclassification of each histotype demonstrated a decisive role in the prognostic stratification. *BRCA*mut HGSCs show better prognosis compared to *BRCA*wt tumors, particularly after the introduction of PARPi in maintenance treatment [39]. *BRAF*^V600E^ LGSC are characterized by better outcomes compared to *KRAS*-mutated LGSCs [55]. ENOCs encompass at least four different molecular subgroups: *POLE*-ultramutated, MSI hypermutated, CN^high^ and CN^low^ tumors with different histologic and clinical features [16]. *POLE*-ultramutated tumors have demonstrated an excellent prognosis, MSI and CN^low^ intermediate outcomes and CN^high^ tumors poor survival rates. Almost 25% of ENOCs, 20% of MCs, and 14% of CCCs could be related to Lynch Syndrome with notable genetic and clinical implications, such as the adoption of preventive familiar strategies and, in the event of adjuvant therapy, of potential checkpoint inhibition [96]. The need to implement worldwide, easy-to-use and cost-effective strategies to identify these features led to the development of the molecular surrogate method, with IHC as the main focus with excellent results (Table 5). Although not as reliable as molecular analysis, in order to avoid unnecessary costly and distressful testing, IHC could represent the upfront standard approach of a pathology laboratory.

## Figures and Tables

**Figure 1 diagnostics-11-00199-f001:**
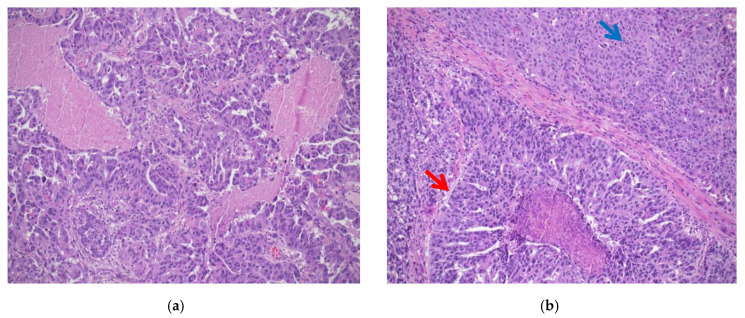
(**a**) High-grade serous carcinoma, classic histology. Papillary and micropapillary architecture with marked cytological atypia, atypical mitoses, and tumor necrosis (hematoxylin-eosin; ×100); (**b**) high-grade serous carcinoma, SET variant, with pseudoglandular (red arrow) and solid (blue arrow) architectural patterns (hematoxylin-eosin; ×100).

**Figure 2 diagnostics-11-00199-f002:**
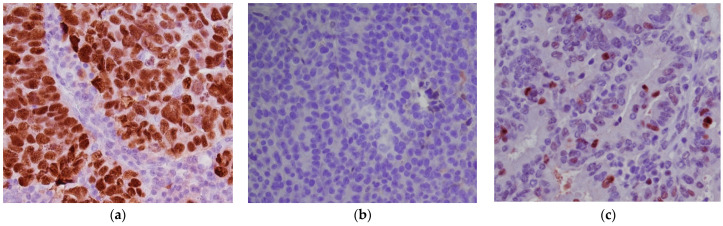
High-grade serous carcinoma, p53 abnormal immunohistochemical patterns. (**a**) p53 overexpression (DO7 clone; ×400); (**b**) p53 complete absence of expression (null pattern) (DO7 clone; ×400); (**c**) p53 wild-type pattern (DO7 clone; ×400).

**Figure 3 diagnostics-11-00199-f003:**
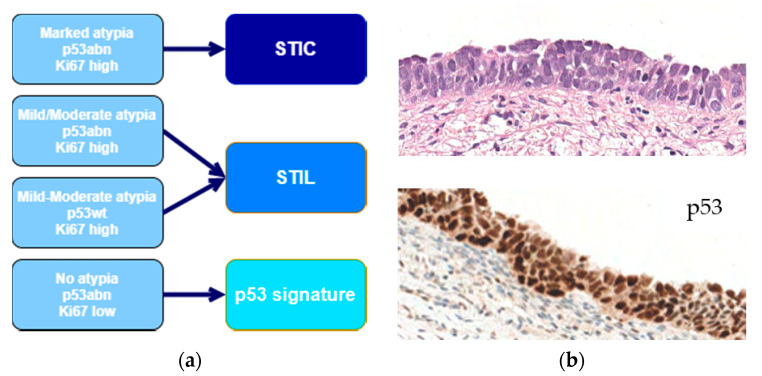
(**a**) Diagnostic algorithm to classify tubal lesions (adapted from R. Vang et al.). abn: abnormal; STIC: serous tubal intraepithelial carcinoma; STIL: serous tubal intraepithelial lesion; wt: wild type. (**b**) An example of STIC (top: marked nuclear atypia, hematoxylin-eosin, ×400; bottom: p53-overexpression, DO7 clone, ×400).

**Figure 4 diagnostics-11-00199-f004:**
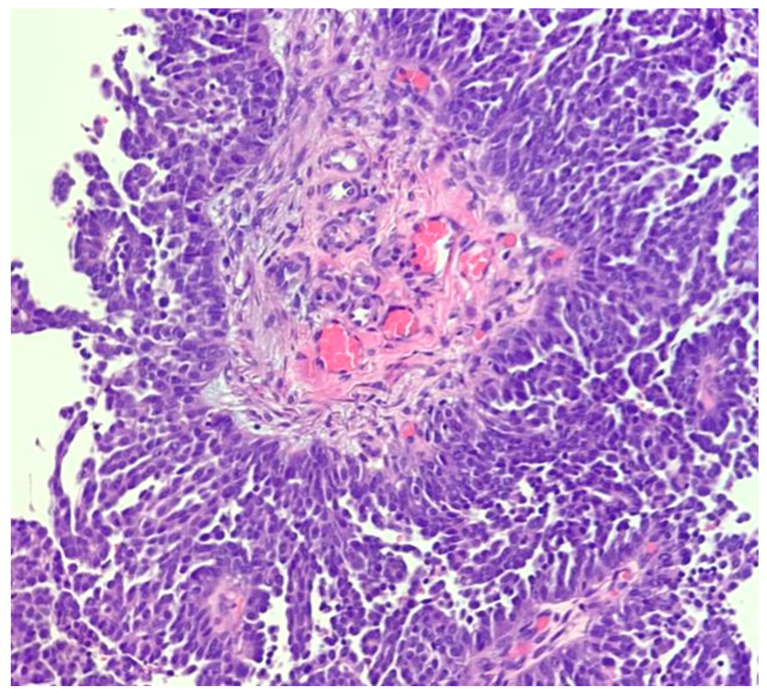
Low-grade serous carcinoma, micropapillary (medusa-like) architecture (hematoxylin-eosin; ×100).

**Figure 5 diagnostics-11-00199-f005:**
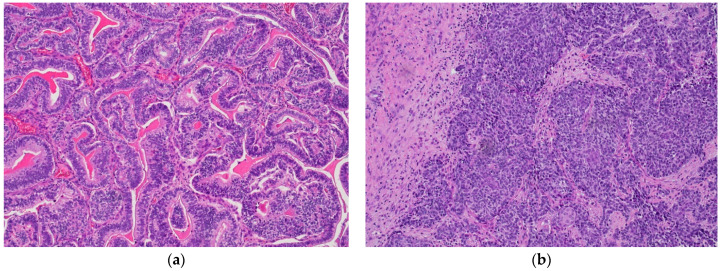
(**a**) Endometrioid ovarian carcinoma, low-grade: back-to-back glands without intervening stroma (hematoxylin-eosin; ×100); (**b**) endometrioid ovarian carcinoma, high-grade, solid growth pattern (hematoxylin-eosin; ×100).

**Figure 6 diagnostics-11-00199-f006:**
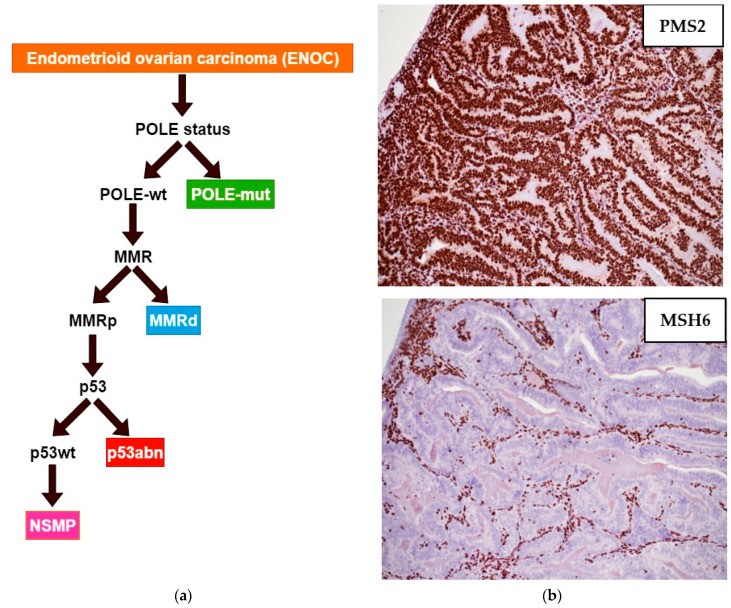
(**a**) Diagnostic algorithm to classify endometrioid ovarian carcinomas (left) (based on P. Krämer et al.). abn: abnormal; MMR: mismatch repair; MMRd: mismatch repair deficient; MMRp: mismatch repair proficient; mut: mutated; NSMP: no specific molecular profile; *POLE*: polymerase-epsilon; wt: wild-type. (**b**) MMRd endometrioid carcinoma low-grade, notable loss of MSH6 with intact PMS2 expression (×100). As internal control, tumor infiltrating lymphocytes and intertumoral stromal cells demonstrated intact MMR expression (×100).

**Figure 7 diagnostics-11-00199-f007:**
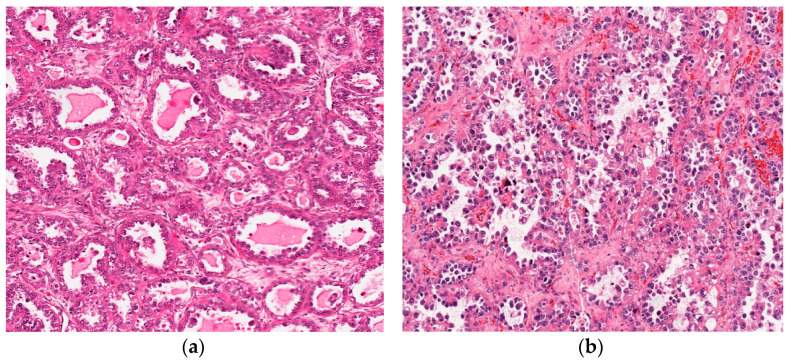
(**a**) Clear cell carcinoma with tubulocystic pattern (hematoxylin-eosin; ×200). (**b**) Clear cell carcinoma with papillary/micropapillary growth pattern (hematoxylin-eosin; ×200).

**Table 1 diagnostics-11-00199-t001:** Molecular markers of high-grade serous carcinoma.

Molecular Markers	Clinical Significance	Immunohistochemistry Available/ Currently Used in Clinical Practice
*TP53*	Found in >95% HGSCs	Yes/Yes *
*BRCA1/2*	Better prognosis, eligible for PARPi therapy	Yes/No
PD1/PD-L1	Better prognosis in late-stage HGSCs	Yes/No

* only used for diagnostic purpose; HGSC: high-grade serous carcinoma; PARPi: poly(ADP-ribose) polymerase inhibtors.

**Table 2 diagnostics-11-00199-t002:** Molecular markers in low-grade serous carcinomas.

Molecular Markers	Clinical Significance	Immunohistochemistry Available/ Currently Used in Clinical Practice
*BRAF* ^V600E^	Good prognosis, candidate for Trametinib therapy	Yes/Yes
Hormone receptors	Candidate for HRT (only advanced-stage LGSC)	Yes/Yes

*KRAS*	Poor prognosis	No/No

*ERBB2*	Unknown	Yes/No

LGSC: low-grade serous carcinoma; HRT: hormonal replacement therapy.

**Table 3 diagnostics-11-00199-t003:** Molecular classification of endometrioid ovarian carcinomas.

TCGA-EEC Class	Molecular Surrogate	Clinical Significance	Immunohistochemistry Available/ Currently Used in Clinical Practice
Hypermutated	*POLE* sequencing (Sanger)	Excellent prognosis, candidate for checkpoint inhibition	No/Yes
Ultramutated	MSI assay	Intermediate prognosis, candidate for checkpoint inhibition	Yes/Yes
CN-high	*TP53* sequencing	Poor prognosis	Yes/Yes
CN-low	*POLE*/MSI/*TP53* wild-type	Intermediate prognosis	Yes/Yes

CN: copy number; MSI: microsatellite instability; *POLE*: polymerase ε; TCGA-EEC: The Cancer Genome Atlas Network classification for Endometrial Endometrioid Carcinomas.

**Table 4 diagnostics-11-00199-t004:** Other molecular markers associated with endometrioid ovarian carcinomas.

Molecular Markers	Clinical Significance	Immunohistochemistry Available/ Currently Used in Clinical Practice
PD1/PD-L1	Candidate for checkpoint inhibition	Yes/Yes (selected cases)
Hormone receptors	Better prognosis, candidate for HRT	Yes/Yes
*PTEN*	Target for PI3K/AKT inhibition	Yes/No *
*ARID1A*	Target for EZH2 and HDAC inhibition	Yes/No *
*CTNNB1*	Good prognosis	Yes/No
*CDKN2A*	Worse prognosis	Yes/No

* currently used only for diagnostic purpose. HDAC: histone deacetylase; HRT: hormonal replacement therapy.

**Table 5 diagnostics-11-00199-t005:** Overview of molecular markers in ovarian epithelial cancers.

Molecular Markers	Histotype	Immunohistochemistry Surrogate Available	Comments
*TP53*	HGSC, ENOC, CCC, CS	Yes	Mutated in 96% HGSCs Worse prognosis in ENOCs
*BRCA1, BRCA2*	HGSC, CS	Yes	Better prognosis, PARPi eligible
*BRAF*	LGSC	Yes	Lower recurrence rate, better prognosis possibility of targeted therapy in advanced LGSC
*KRAS*	LGSC, MLA, CS	No	Higher recurrence rate, worse prognosis
*ERBB2*	LGSC, MC	Yes	Better prognosis in MC, no data available in LGSC
Hormone receptors	LGSC, ENOC	Yes	Diffuse expression associated with better prognosis, possibility of HRT in LGSC and low-grade ENOC
*MLH1, PMS2, MSH2, MSH6*	ENOC, CCC, MC, DEDC/UC	Yes (MLH1 methylation analysis required for MLH1/PMS2 loss)	Better prognosis, 80% cases due to somatic MLH1 hypermethylation, germline mutations associated with Lynch Syndrome, high PD-L1 expression (checkpoint inhibition candidate)
*POLE*	ENOC, CCC, DEDC/UC	No	Excellent prognosis, high PD-L1 expression (checkpoint inhibition candidate)

HGSC: high-grade serous carcinoma; LGSC: low-grade serous carcinoma; ENOC: endometrioid ovarian carcinoma; CCC: clear cell carcinoma; MC: mucinous carcinoma; MLA: mesonephric-like adenocarcinoma; DEDC/UC: dedifferentiated/undifferentiated carcinoma; CS: carcinosarcoma; HRT: hormonal replacement therapy.

## Data Availability

Not applicable.

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
