# Peer review of "Immunohistochemical Biomarkers as a Surrogate of Molecular Analysis in Ovarian Carcinomas: A Review of the Literature"

_diagnostics, 2021, doi:10.3390/diagnostics11020199_

Round 1

Reviewer 1 Report

This work by Santandrea et al. is a very interesting paper, well planned and well written, encompassing a large review in a crucial issue in the diagnosis of ovarian cancers.

In this Reviewer's opinion the paper is worthy of consideration, but it still needs minor revisions.

(lines 61,66): why if there are 2 figs 1 (a,b), in the text there is no mention for them, but just a generic fig 1 cross reference?

(lines 83,85): again; why if there are 3 figs 2 (a,b,c), in the text there is no mention for all of them, but just two generic fig 2 cross references?

(line 85) Furthermore, at line 85, in this Reviewer's opinion, fig 3 is a typo and it should be modified in fig 2(c).

(line 98) STIC, although it is mentioned for the first time, it is written as acronym. 

(line 114) AJCC is worthy of reference (as it is listed in ref section)

Starting line 137, please, the Authors should pay great attention to the references sequence of citations in the text (or otherwise, should shorten the list of references). In fact, there are some references listed but never cited in the text; furthermore, many times the reference order in the text is not strictly followed.

For example: line 136 reff 21-24; then the following ref is the number 31 at line 146, lacking reff 29 and 30 which seem to be never cited. The same for ref 38 never cited and many other reffs, including the last one 117..

Ref 40 (line 158) cited before ref 39 (line 161) and so on for reff 50,51 apparently never cited and many other similar cases of misponed refs.

Line 238: fig 5 has the same problems of figs 1 and 2; moreover, in the tex Authors discuss about architectural pattern, nuclear atypia, tumor grade and upgrading tumor, but in the caption there is no mention at all of all these features. Finally, in this Reviewer's opinion, the quality of the two pictures is very poor: as they are, they are useless for a reader who want to appreciate the cytological details.

Lines 241 and 331 Progesterone Receptor is written as PR, while before had been used PGR: please, Authors are encouraged to conform.

Table 3 and Figure 6 should be moved in more correct positions. 

Lines 309, 331, 333 the year of publication is not necessary.

Author Response

We thank the reviewer for the careful reading of the manuscript and the constructive remarks. We have taken the comments on board to improve and clarify the manuscript. Please find below a detailed point-by-point response to all comments

Point 1:  (lines 61,66): why if there are 2 figs 1 (a,b), in the text there is no mention for them, but just a generic fig 1 cross reference?

Response 2: Figures 1a and 1b have been now mentioned in the main text.

Point 2: (lines 83,85): again; why if there are 3 figs 2 (a,b,c), in the text there is no mention for all of them, but just two generic fig 2 cross references?

Response 2: Also figures 2a, 2b and 2c have been now mentioned in the main text.

Point 3: (line 85) Furthermore, at line 85, in this Reviewer's opinion, fig 3 is a typo and it should be modified in fig 2(c).

Response 3: the typo has been corrected.

Point 4: (line 98) STIC, although it is mentioned for the first time, it is written as acronym

Response 4: STIC is now written as "Serous Tubal Intraepithelial Carcinoma".

Point 5: (line 114) AJCC is worthy of reference (as it is listed in ref section)

Response 5: AJCC is now mentioned as a reference.

Point 6: Starting line 137, please, the Authors should pay great attention to the references sequence of citations in the text (or otherwise, should shorten the list of references). In fact, there are some references listed but never cited in the text; furthermore, many times the reference order in the text is not strictly followed.

For example: line 136 reff 21-24; then the following ref is the number 31 at line 146, lacking reff 29 and 30 which seem to be never cited. The same for ref 38 never cited and many other reffs, including the last one 117..

Ref 40 (line 158) cited before ref 39 (line 161) and so on for reff 50,51 apparently never cited and many other similar cases of misponed refs.

Response 6: every reference has been reviewed and it should be now correctly mentioned in the main text

Point 7: Line 238: fig 5 has the same problems of figs 1 and 2; moreover, in the tex Authors discuss about architectural pattern, nuclear atypia, tumor grade and upgrading tumor, but in the caption there is no mention at all of all these features. Finally, in this Reviewer's opinion, the quality of the two pictures is very poor: as they are, they are useless for a reader who want to appreciate the cytological details.

Response 7: figure 5a and 5b are now mentioned in the main text. We apologize for the poor quality of the pictures, according to the editors the images have been significantly compressed in the uploading process. It should now be fixed.

Point 8: Lines 241 and 331 Progesterone Receptor is written as PR, while before had been used PGR: please, Authors are encouraged to conform.

Response 8: PGR acronym is now used across the entire manuscript.

Point 9: Table 3 and Figure 6 should be moved in more correct positions. 

Response 9: They should be now in the respective correct positions

Point 10: Lines 309, 331, 333 the year of publication is not necessary.

Response 10: year of publication has been removed.

Reviewer 2 Report

Santandrea and collaborators perform a review on the role of immunohistochemistry, as a surrogate of molecular analysis, on the everyday diagnostic approach of ovarian carcinoma. The paper is well written and with interest for the entire community.

Minor concerns:

General comments:

All the figures are presented with different magnifications if possible, the figures should have the same magnification.

On Figure 1 the pseudoglandular and solid architectural patterns should be pointed out on the figure.

Figure 2. The magnification of the figures should be increased in order to have a better view of the p53 pattern staining.

The sentence on the line 134 to141 needs to be reformulated  

When the authors mention the potential use of upfront, pathology-based, BRCA identification is particularly interesting for the identification of the patient with somatic mutations is difficult to understand the real application of this approach with the recent advances in the molecular diagnostics already implemented into the clinic where patients are already tested for the BRCA1 and BRCA2 status by targeted DNA sequencing.

Line 344-346 should be revisit

Author Response

We thank the reviewer for the careful reading of the manuscript and the constructive remarks. We have taken the comments on board to improve and clarify the manuscript. Please find below a detailed point-by-point response to all comments.

Point 1: All the figures are presented with different magnifications if possible, the figures should have the same magnification.

Response 1: The figures have now the same magnification. However for Clear Cell Carcinoma we kept a slightly higher magnification to better illustrate the architectural patterns.

Point 2: On Figure 1 the pseudoglandular and solid architectural patterns should be pointed out on the figure.

Response 2: we have now pointed the solid and pseudoglandular patterns with red and blue arrows.

Point 3: Figure 2. The magnification of the figures should be increased in order to have a better view of the p53 pattern staining.

Response 3: The magnification have been increased.

Point 4: The sentence on the line 134 to 141 needs to be reformulated.

Response 4: We reformulated the sentence.

Point 5: When the authors mention the potential use of upfront, pathology-based, BRCA identification is particularly interesting for the identification of the patient with somatic mutations is difficult to understand the real application of this approach with the recent advances in the molecular diagnostics already implemented into the clinic where patients are already tested for the BRCA1 and BRCA2 status by targeted DNA sequencing.

Response 5: we thank the reviewer for  the constructive remarks. What we tried to highlight is that in some countries the patients is still sent to a genetical referral regardless of the histological diagnosis, the BRCA-reflex testing could therefore avoid unnecessary and distressful genetic counseling. Furthermore, in the near future, BRCA-reflex testing could be further restricted to selected ovarian cancer histotypes (such as HGSC) to be even more cost-effective.

Point 6: Line 344-346 should be revisit

Response 6: the lines have been revisited.